# Keep It Real: Challenges in Attacking Compression-Based Adversarial Purification

**Samuel Räber**
ETH Zürich
Switzerland
sraeber@ethz.ch

**Till Aczel**
ETH Zürich
Switzerland
taczel@ethz.ch

**Andreas Plesner**
ETH Zürich
Switzerland
aplesner@ethz.ch

**Roger Wattenhofer**
ETH Zürich
Switzerland
wattenhofer@ethz.ch

## Abstract

Previous work has suggested that preprocessing images through lossy compression can defend against adversarial perturbations, but comprehensive attack evaluations have been lacking. In this paper, we construct strong white-box and adaptive attacks against various compression models and identify a critical challenge for attackers: high realism in reconstructed images significantly increases attack difficulty. Through rigorous evaluation across multiple attack scenarios, we demonstrate that compression models capable of producing realistic, high-fidelity reconstructions are substantially more resistant to our attacks. In contrast, low-realism compression models can be broken. Our analysis reveals that this is not due to gradient masking. Rather, realistic reconstructions maintaining distributional alignment with natural images seem to offer inherent robustness. This work highlights a significant obstacle for future adversarial attacks and suggests that developing more effective techniques to overcome realism represents an essential challenge for comprehensive security evaluation.

## 1 Introduction

Adversarial attacks on image classification models involve making small perturbations to an image such that the classifier's output changes–even though the image appears semantically unchanged to a human observer. A model can be trained to be robust against such adversarial examples, for example, by augmenting the training data with noise or showing the model adversarial examples during training. Another strategy is applying a transformation to the input image that preserves its semantic content while altering it to render the adversarial noise ineffective. This approach has the advantage of being able to be used for any classification model without requiring retraining.

Lossy image compression often discards details deemed perceptually unimportant, which may include the subtle perturbations introduced by adversarial attacks. Early work argued that standard

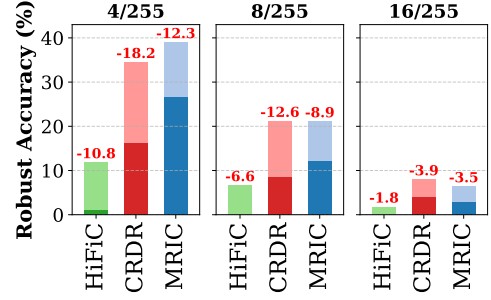

Figure 1: Decrease in robust accuracy when employing a compression defense with reduced realism under different perturbation budgets. Incorporating realism substantially increases the difficulty of successful attacks.

codecs like JPEG yield modest robustness gains [1], though they often introduce compression artifacts that push images outside the classifier's training distribution. More recently, learned compression

39th Conference on Neural Information Processing Systems (NeurIPS 2025) Workshop: Reliable ML from Unreliable Data.

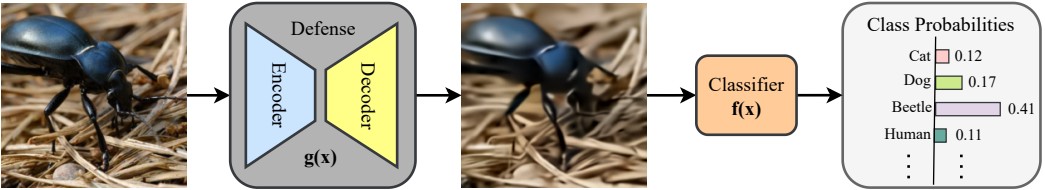

Figure 2: Overview of compression-based adversarial defense. An input image (potentially containing adversarial perturbations) is first processed by the defense module, which consists of an encoder-decoder architecture that compresses and reconstructs the image. This reconstructed image is then passed to a classifier, which outputs class probabilities. The defense aims to indirectly, through the compression process, remove adversarial noise while preserving the semantic content needed for correct classification.

models have promised to bridge this gap by generating visually plausible reconstructions that may also remove stronger adversarial noise [2].

However, many proposed defenses—especially preprocessing-based ones—have been criticized for relying on gradient masking; they make general gradient-based attacks harder without genuinely increasing robustness [3]. When attackers adapt to circumvent gradient obfuscation (e.g., by approximating or smoothing gradients), many defenses fail [4–7]. This raises two critical questions:
(1) *Do robustness gains from image compression persist under rigorous, adaptive attacks?*
(2) *If so, what underlying mechanism contributes to this robustness?*

We argue that realism in reconstructed images is a key factor underpinning the robustness. Our results show that only compression models capable of producing high-fidelity, realistic images offer meaningful robustness against strong adaptive adversarial attacks, while other compression models can be broken. Realism aids robustness in two ways: It avoids unnatural artifacts that shift images off-distribution, and by hallucinating in semantically plausible details that obscure adversarial noise.

A highly realistic compression model reconstructs images close to the original and free from perceptible signs of compression. Classifier models often perform poorly on out-of-distribution inputs; ensuring realism in reconstructed images helps keep them within the distribution of natural images. High realism can be achieved by hallucinating in plausible details. For example, while the exact texture of tree leaves may be removed during compression, a realistic model will hallucinate plausible leaf-like textures. These added details can help obscure adversarial noise, making it more difficult for an attacker to craft successful perturbations.

In this paper, we systematically evaluate the robustness of compression-based defenses and isolate the role of realism. Our analysis builds on the findings in [2], where it was argued that human-aligned compression contributes to robustness. We re-evaluate their claims under rigorous adversarial threat models and identify shortcomings in their evaluation protocol. We show that their observed robustness stemmed not merely from compression, but specifically from realism. We reinforce this conclusion through extensive evaluation across a broader and newer set of learned compression models.

## 2 Background and Related Works

Our work lies at the intersection of two big fields, and we aim to give complete overviews of both. However, due to the page limit, we had to move the complete related work to Appendix B.

### 2.1 Adversarial Robustness

Shortly after the success of AlexNet [8], it was found that neural networks are very susceptible to *adversarial attacks* [9, 10]. Here, an adversary adds (usually) small and imperceptible perturbations to an image such that a model mislabels it.

**Attacks** Many attacks have been developed over the years with various benefits and drawbacks. Some of the most noteworthy are FGSM (Fast Gradient Sign Method) [10], iterated FGSM or iFGSM [11], CW [12], and PGD (Projected Gradient Descent) [13]. These attacks fall into two categories:

FGSM, iFGSM, and PGD are $l_\infty$-bounded attacks, and CW is an $l_2$-bounded attack. We refer the reader to the original papers for specific details, but include some details in Appendix C. The TLDR is: PGD has a perturbation budget $\epsilon$ and an iteration budget $n$. PGD does projected gradient descent for $n$ iterations to find an adversarial example at most $\epsilon$ distance in $l_\infty$ from the original image.

When the defense includes randomness, a common attack augmentation is EoT (Expectation over Transformation), where gradients are averaged over multiple backward passes [14, 7]. If the defense causes gradient masking, Athalye et al. [14] suggested Backward Pass Differentiable Approximation (BPDA), where the defense is used during the forward pass, but a differentiable approximation is used during the backward pass. The simplest is to use the identity function, while more advanced methods might train a differentiable surrogate model $g'$ that emulates the defense $g$; i.e., $g(x) \approx g'(x)$.

Lastly, often the best attack results are found using *adaptive attacks* [4–7]. Here, the attacks, for instance, the optimization objective, are adjusted to the defense. This work focuses on adaptive attacks with PGD as the underlying optimization method.

**Defenses**  Given the prevalence of adversarial attacks, many researchers have explored how to defend against them. We broadly view this in three groups: Architecture improvements [15–17], adversarial training [9, 10, 13, 18, 17], and adversarial purification [19–21, 2]. The architecture branch focuses on making the models more robust by design, for instance, by taking inspiration from biology to mimic the human eye when training a ResNet model [17]. Adversarial training consists of showing adversarial examples during training to ensure correct classifications. This has yielded positive results [9, 13], but models can be broken [7].

Adversarial purification aims to remove the adversarial noise in images before passing the cleaned images to pretrained classification models. It is motivated by the work of Ilyas et al. [22], hinting that adversarial examples perturb brittle features in the model. These methods *should work independently* of any robustness applied to the classifier through adversarial training or robust architectures.

Diffusion models have been proposed as a way to remove adversarial noise from input images [19, 23]. These approaches are conceptually similar to compression-based defenses with realism: both aim to project adversarial examples back onto the manifold of natural images to restore classifier performance. However, prior work on diffusion-based purification has not explicitly investigated the role of realism as a contributing factor to robustness. One major drawback of diffusion-based defenses is their computational cost [24, 25]. But simply making gradients difficult to compute does not equate to genuine robustness. The purification method proposed by Nie et al. [19] was later defeated by Lee and Kim [23]. However, the latter only evaluated their improved defense under the attack that broke the former. They did not develop new adaptive attacks tailored to their defense—a strategy that past research suggests would likely reduce the effectiveness of the defense. The history of adversarial robustness research shows that defenses that are not tested under strong, tailored attacks often overstate their robustness [14, 4, 5].

### 2.2 Realism in Image Compression

Image compression algorithms are traditionally evaluated using *distortion metrics*, which measure the distance between a restored image $\hat{x}$ and its reference $x$. Formally, distortion is defined as:

$$\mathcal{D} := \mathbb{E}_{(x,\hat{x}) \sim (p_X, p_{\hat{X}})}[\Delta(x, \hat{x})], \tag{1}$$

where $\Delta(\cdot, \cdot)$ is a pointwise distortion measure (e.g., $\ell_2$ distance), and $p_X$, $p_{\hat{X}}$ denote the distributions of ground truth and reconstructed images. Distortion is considered a *full-reference* metric, requiring access to the original image $x$ to compare it to the reconstructed version $\hat{x}$.

However, metrics such as PSNR, MS-SSIM [26], or LPIPS [27] correlate poorly with human perception, and directly quantifying perceptual distortion remains a challenging problem. Instead of a perceptual distortion metric, one can measure both distortion and realism and optimize the compression model for both. Realism can be formally defined as:

$$\mathcal{R} := -d(p_{\hat{X}}, p_X), \tag{2}$$

where $d(\cdot, \cdot)$ is a divergence measure, such as Kullback-Leibler. Unlike distortion, realism is a *no-reference metric*, requiring only the generated image distribution to match that of natural images. Although measuring realism remains challenging [28], a widely used proxy is the Fréchet Inception

Distance (FID) [29], which compares the distributions extracted by a pretrained Inception network. FID has gained popularity for capturing both fidelity and diversity in generated samples.

Compression models are typically trained with the following loss function:

$$\mathcal{L} = \mathcal{L}_{\text{RATE}} + \lambda \mathcal{D} - \beta \mathcal{R}, \tag{3}$$

where $\mathcal{L}_{\text{RATE}}$ represents the estimated rate, or in other words, the number of bits required to represent the image after it has been compressed, $\lambda$ controls the level of distortion, and $\beta$ the level of realism. Since the information is not transmitted through an explicit information bottleneck in our approach, the rate term $\mathcal{L}_{\text{RATE}}$ does not play a critical role in this context.

Compression models can be broadly categorized into those optimized for distortion (e.g., JPEG [30], Hyperprior [31], ELIC [32]) and those explicitly trained to maximize realism (e.g., HiFiC [33], PO-ELIC [34], WD [35], ConHa [36]). More recent approaches, such as MRIC [37] and CRDR [38], provide control over the realism–distortion tradeoff within a single network by conditioning on $\lambda$ and $\beta$. This controllability enables direct investigation of how realism influences adversarial robustness.

Our work builds on this foundation, exploring the intersection of realistic compression and adversarial robustness. We extend the existing literature by providing experimental evidence that realism, rather than distortion, makes image-compression models (partially) robust against adversarial examples.

### 2.3 Compression as Adversarial Defense

Using compression as a defense for neural networks is not a new idea. It has been explored for almost a decade [1, 39–42] where some authors also explored using iterated compression and decompression cycles [42, 2]. Two key works in this area are by Guo et al. [1] and Shin and Song [3]; the former argued that JPEG compression as a preprocessing step is a very effective adversarial defense while the latter showed that, by making JPEG differentiable, this defense could be bypassed entirely–highlighting the necessity of properly evaluating a defense.

## 3 Methodology

Our work evaluates the compression models' robustness; thus, we focus on the ImageNet classification benchmark, a well-established benchmark with high-resolution images. This gives us many options for pretrained models, allowing a more exhaustive model evaluation. If otherwise not stated, we use the full validation split of the ImageNet dataset (50000 images). We used two different classification models, ResNet50 [43] and ViT B 16 [44]. For PyTorch models, we use the improved weights `IMAGENET1K_V2` for ResNet50 and the default weights `IMAGENET1K_V1` for ViT B 16. For TensorFlow models, we use the default ResNet50 pretrained weights `imagenet`. Certain ablations are only done for ResNet to reduce the compute load; the results have only minor differences to ViT.

### 3.1 Defenses

In the evaluated defense strategy, we integrate a compression and decompression step into the image classification pipeline (cf. Figure 2). The image compression model acts as a preprocessing step, transforming the image before classification. The transformation discards certain information from the image by employing lossy compression techniques. This process can mitigate the effect of adversarial perturbation on the classifier's predictions, thereby enhancing the model's robustness.

As we demonstrate later, realism plays a crucial role in the robustness of this pipeline. Therefore, we focus our experiments on models that either explicitly control the level of realism or exist in both standard and enhanced realism variants. In particular, MRIC [37] and CRDR [38] offer variable realism settings, and we denote their low- and high-realism variants as *MRIC LR*, *MRIC HR*, *CRDR LR*, and *CRDR HR*, respectively. We also consider models available in both standard rate-distortion and rate-distortion-realism versions, such as *Hyperprior* [45] and *HiFiC* [33]. To further validate the importance of realism, we include *JPEG* [30] and *ELIC* [32] (noting that the high-realism version, PO-ELIC [34], does not have publicly available pretrained weights), and show that these consistently underperform compared to high-realism compression models. For white-box attacks, we require access to gradients; for learned compression models, gradient computation is natively supported, while for JPEG, we employ a continuous relaxation approach [3]. We focus on VAE-based methods as diffusion or INR-based approaches are prohibitively expensive to evaluate at scale [19, 23].

## 3.2   Threat models

We define our own threat models that encompass the prior experiments from Räber et al. [2] to ensure proper evaluation [46, 4, 47]. The threat models follow standard formulations from Biggio et al. [48], Carlini and Wagner [46], Pang et al. [49]. Specifically, we consider $l_\infty$ untargeted attacks with perturbation budget $\epsilon$, i.e., for an original image $x$ and perturbed image $x'$, we have $\|x - x'\|_\infty \leq \epsilon$.

Our primary focus is on PGD attacks, widely regarded as among the strongest given sufficient objective formulation and computational resources (see Appendix C). We also include adaptive attacks, where the adversary, aware of the defense mechanism, tailors the attack accordingly. Adaptation in this context means the adversary can specialize the attack by studying the defense and looking for ways to defeat it [5]. These adaptive attacks still use PGD (see Section 3.3 for details). If otherwise not stated, we use 10 PGD iterations.

Lastly, we consider three adversary knowledge levels as described in Carlini and Wagner [46]. **Black-box (BB)**: The adversary does not know the defense or its existence. The attacker can create adversarial perturbations with respect to the gradients of the classifier or the outputs of the classifier. **Gray-box (GB)**: The adversary knows the defense is present and can use it for the forward pass only; they cannot compute gradients through the defense. This threat variant is used for adaptive attacks. **White-box (WB)**: The adversary has complete knowledge and access to the defense. The gradients of the defense and the output of the combined defense and classifier pipeline are used in the attack.

## 3.3   Adaptive Attacks Against Compression Defenses

A common pitfall in adversarial robustness papers is the lack of honest effort in implementing proper attacks against one's own defense [4–6]. In our evaluation, we conduct adaptive attacks on all compression models, which means the attacks are tailored to exploit the target model's weaknesses. This section reviews the adaptive attacks used against the compression defenses. We use the following notation: $f$ is the image classifier, $g$ is a compression defense (compression and decompression), and $h = f \circ g$. $x$ is an image with label $y$, and $L$ is a loss function. We use cross-entropy as the loss function for all our experiments (except ACM). $\nabla_x f$ is the gradient of $f(x)$ with respect to $x$.

**ST BPDA**   For the first adaptive attack, we use BPDA with the straight through (ST) approximation; this assumes $g(x) \approx x$. For the forward pass, the defense is used, but the straight-through (ST) estimator is applied in the backward pass, replacing the compression model's gradient with the identity function. Thus, $\nabla_x h \coloneqq \nabla_x f(x)|_{x=g(x)}$.

**U-Net BPDA**   In the second adaptive attack, we use BPDA with a standard U-Net [50] trained to approximate the compression defense $g$ (see training details in Appendix F). The idea is that if $g$ primarily causes gradient masking, then replacing it with a differentiable proxy $g'$ should yield meaningful gradients for the attack. During the forward pass, we use the actual defense, $h(x) = f(g(x))$, while the backward pass substitutes $g$ with $g'$, yielding gradients $\nabla_x h \coloneqq \nabla_x (f \circ g')(x)$.

**Attacks on the Compression Model (ACM)**   The third adaptive attack focuses only on attacking the compression method. Instead of focusing on misclassifications in the classifier through the cross-entropy loss, the attacker uses the objective function $MSE(x, g(x))$ for the attack. The goal is then to cause significant distortions that confuse the classifier.

**Adaptive Realism Attack (ARA)**   The fourth adaptive attack is specific to models with varying realism. Let $g_\beta$ be a defense with realism parameter $\beta$, the goal of the attack is then for a given $\beta$ to find the $\beta'$ that gives the lowest model accuracy when we attack $f(g_\beta(x))$ with $\nabla_x h \coloneqq \nabla_x (f \circ g_{\beta'})$.

**Considerations for the Attacks**   All the VAE-based compression defenses are deterministic; thus, EoT [51] should not be required for these defenses [14].

Adaptive attacks aim to craft adversarial examples explicitly tailored to the defense mechanism—in our case, the compression model. White-box PGD (WB PGD) and U-Net BPDA perform best in our evaluated attacks. The primary difference between WB PGD and U-Net BPDA is that U-Net BPDA approximates the backward pass of the compression model with the gradients of U-Net trained to mimic the outputs of the compression model, allowing gradients to flow more freely.

| Classifier | Defense | 4/255 | | 8/255 | | 16/255 | |
|---|---|---|---|---|---|---|---|
| | | LR | HR | LR | HR | LR | HR |
| RESNET | HYPERPRIOR | 0.98 | **11.83** | 0.02 | **6.65** | 0.00 | **1.80** |
| | MRIC | 26.68 | **39.00** | 12.20 | **21.10** | 2.90 | **6.40** |
| | CRDR | 16.30 | **34.50** | 8.60 | **21.18** | 4.10 | **7.98** |
| | JPEG | 5.19 | — | 0.24 | — | 0.01 | — |
| | ELIC | 16.43 | — | 4.98 | — | 0.34 | — |
| VIT | HYPERPRIOR | 0.20 | **10.06** | 0.00 | **2.64** | 0.00 | **0.16** |
| | CRDR | 14.44 | **28.28** | 6.98 | **14.90** | 2.16 | **2.46** |
| | JPEG | 0.88 | — | 0.04 | — | 0.00 | — |
| | ELIC | 14.12 | — | 2.38 | — | 0.10 | — |

Table 1: ResNet accuracy under adaptive attacks on compression models for ImageNet. Hyperprior and HiFiC results are combined to give the low and high realism. JPEG and ELIC do not have a high realism (HR) version; only low realism (LR) is available. Note, for MRIC, only the PGD attack was applied. We selected each architecture's most effective adaptive attack based on evaluations in Table 3 and Table 6. We see that HR models consistently perform better.

## 4 Experiments

We present a series of experiments to support our claim that realism makes compression-based defenses hard to attack. First, we demonstrate that defenses incorporating realism consistently outperform those that do not (Section 4.1). Next, we verify that the observed robustness does not arise from gradient masking (Section 4.2). Finally, we construct stronger model-specific adaptive attacks, showing that even under these conditions, high-realism defenses remain more robust (Section 4.3).

### 4.1 Role of Realism

Table 1 shows that models incorporating realism into their reconstructions consistently achieve higher robust accuracy under strong adaptive attacks. These results provide compelling empirical support for our central hypothesis: *realism plays a key role in reducing the effectiveness of adversarial attacks*. ViT models exhibit lower overall robustness than ResNets across the board.

To isolate the impact of realism, we analyze two compression models where realism can be explicitly controlled. As shown in Figure 6 (Appendix G.2), increasing realism in reconstructed images monotonically improves robustness. While distortion has been the focus of extensive prior work [39, 3, 2], realism remains largely unexplored as a factor in defense performance. Our results highlight a key difference: distortion presents an inherent trade-off. If the distortion is too low, adversarial noise is preserved. If the distortion is too high, the reconstruction discards critical semantic information, making the image unrecognizable. In both cases, the defense fails—either by retaining harmful perturbations or by producing images that are implausible under the natural data distribution. In contrast, increasing realism consistently improves robustness without the trade-offs typically associated with distortion. Prior to this work, the role of realism in adversarial defenses had not been thoroughly investigated.

These findings suggest that compression alone is insufficient as a defense mechanism. Without realism, compression may eliminate adversarial perturbations but also introduce artifacts that push reconstructions off the natural data manifold. Such off-distribution reconstructions can make downstream classifiers more vulnerable to attack. Realistic reconstructions, by contrast, preserve semantic content while introducing plausible details, helping keep inputs within the natural data distribution and more effectively mask adversarial signals.

### 4.2 Gradient Masking

Modifying the attack budget $\epsilon$ does not influence gradient masking as it merely changes the size of the space in which attacks can search for adversarial examples. Therefore, if the model consistently fails under higher $\epsilon$ values, the robustness observed at smaller $\epsilon$ values is more likely to reflect genuine defense rather than gradient obfuscation. In Table 2, we evaluate CRDR with both low and high realism settings under a range of $\epsilon$ values and PGD iteration counts. Under a 400-step PGD

| ITERATIONS | DEFENSE | STANDARD | 2/255 | 4/255 | 8/255 | 16/255 |
|---|---|---|---|---|---|---|
| 400 | HYPERPRIOR | 79.20 | 79.06 | 78.92 | 79.18 | 77.84 |
| | HYPERPRIOR N | 79.20 | 79.10 | 79.22 | 78.74 | 76.58 |
| | JPEG | 71.02 | 1.44 | 0.26 | 0.02 | 0.02 |
| | CRDR LR | 46.14 | 19.60 | 7.68 | 1.52 | 0.26 |
| | CRDR HR | 62.02 | 36.92 | 19.22 | 6.46 | 1.30 |
| 200 | CRDR LR | 46.14 | 19.52 | 7.70 | 1.70 | 0.28 |
| | CRDR HR | 62.02 | 36.70 | 19.94 | 7.46 | 1.84 |
| 100 | CRDR LR | 46.14 | 20.14 | 8.24 | 2.42 | 0.38 |
| | CRDR HR | 62.02 | 37.12 | 20.84 | 8.86 | 2.54 |
| 50 | CRDR LR | 46.14 | 20.44 | 9.38 | 3.12 | 0.70 |
| | CRDR HR | 62.02 | 38.08 | 23.30 | 10.86 | 4.06 |
| 10 | CRDR LR | 46.14 | 26.40 | 16.24 | 8.56 | 3.84 |
| | CRDR HR | 62.02 | 46.08 | 36.10 | 24.36 | 13.86 |

Table 2: WB PGD attacks with varying numbers of iterations. Only 5000 samples were used due to the increased computational cost. *Hyperprior Noise* refers to a defense in which the gradients through the hyperprior are replaced with a random vector, showing that it suffers from gradient masking.

| STRENGTH | DEFENSE | STANDARD | BB PGD | WB PGD | ST BPDA | U-NET BPDA | ACM | ARA |
|---|---|---|---|---|---|---|---|---|
| 4/255 | HYPERPRIOR | 78.73 | 48.76 | 78.84 | 10.94 | **0.98** | 78.82 | — |
| | HIFIC | 61.53 | 59.52 | **11.83** | 44.65 | 24.04 | 59.20 | — |
| | MRIC LR | 52.80 | 51.04 | **26.68** | — | — | — | **26.68** |
| | MRIC HR | 63.06 | 59.28 | **39.00** | — | — | — | 39.60 |
| | CRDR LR | 46.02 | 44.92 | **16.30** | 39.36 | 28.96 | 41.28 | **16.30** |
| | CRDR HR | 61.72 | 59.80 | 35.88 | 56.36 | 47.62 | 55.67 | **34.50** |
| | JPEG | 70.30 | 65.68 | 8.72 | 18.70 | **5.19** | 68.67 | — |
| | ELIC | 60.03 | 58.79 | **16.43** | 40.00 | 17.98 | 54.48 | — |
| 8/255 | HYPERPRIOR | 78.73 | 27.16 | 78.72 | 3.18 | **0.02** | 78.67 | — |
| | HIFIC | 61.53 | 56.84 | **6.65** | 32.19 | 7.98 | 55.37 | — |
| | MRIC LR | 52.80 | 49.44 | **12.20** | — | — | — | 12.20 |
| | MRIC HR | 63.06 | 57.38 | 21.98 | — | — | — | 21.10 |
| | CRDR LR | 46.02 | 44.14 | **8.60** | 33.58 | 12.12 | 36.96 | **8.60** |
| | CRDR HR | 61.72 | 57.36 | 23.92 | 50.77 | 26.64 | 49.35 | **21.18** |
| | JPEG | 70.30 | 60.76 | 5.19 | 10.01 | **0.24** | 64.75 | — |
| | ELIC | 60.03 | 57.59 | 9.01 | 27.24 | **4.98** | 51.63 | — |

Table 3: Results for ResNet50. "—" denotes values not implemented for MRIC or evaluations invalid without realism control. Models with strong gradient masking, like Hyperprior, are vulnerable to adaptive attacks. Realism does not increase gradient obfuscation, so both high-realism models retain accuracy with minor drops under adaptive attacks. Due to space, the 16/255 are omitted from this table. See Table 5 for these results.

attack with high $\epsilon$, all models fail except for Hyperprior. To investigate this anomaly, we include a "Hyperprior Noise" variant, which replaces Hyperprior's gradient with random noise. Its comparable performance suggests that Hyperprior's apparent robustness is attributable to gradient masking. As the attack budget $\epsilon$ is reduced, the resulting accuracy gains reflect genuine robustness rather than gradient obfuscation. Realism proves critical: at $\epsilon = 4/255$, increasing realism improves CRDR's robust accuracy from 7.68% to 19.22%. This sharp gain underscores realism's essential role in compression-based adversarial defenses.

To complement our quantitative evaluations, in Figure 3 we visualize the loss landscapes for the classifier, CRDR with low realism, and CRDR with high realism, following the implementation described by Zhang et al. [7]. CRDR with low realism exhibits some level of gradient masking, as reflected in the spiky and irregular structure of its loss landscape. Importantly, increasing realism does not increase this effect—the loss landscape remains similarly smooth, suggesting that both low- and high-realism models exhibit comparable levels of gradient masking. This observation is

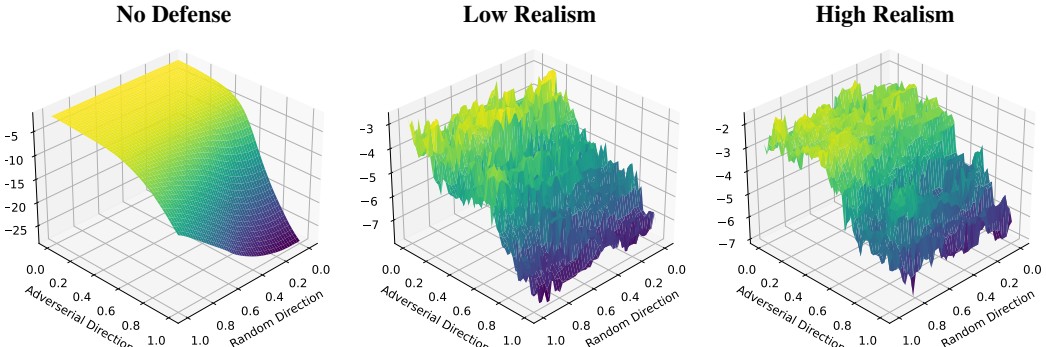

Figure 3: Loss landscapes under successful 100-step PGD attacks on a ResNet with CRDR defense. **Left**: Attacking the classifier directly. **Middle**: Attacking with low realism defense. **Right**: Attacking high realism defense. The standard deviations of the loss surfaces are 0.0544, 0.3343, and 0.3156, respectively. Increasing realism does not make the loss landscape spikier, indicating that it does not contribute to gradient masking.

supported quantitatively: the standard deviations of the loss surfaces for the classifier, low-realism, and high-realism models are 0.0544, 0.3343, and 0.3156, respectively. Despite this, CRDR with high realism consistently outperforms its low-realism counterpart. The loss landscape analysis further supports this conclusion: while both models may exhibit some degree of gradient masking, increased realism does not exacerbate it and plays a direct role in enhancing robustness.

### 4.3 Adaptive Attacks

Many defenses rely on gradient masking, hiding gradients to appear robust rather than truly resisting attacks. This raises the question: Is realism just gradient masking or genuinely robust? To address this, we perform extensive adaptive attacks designed to overcome gradient obfuscation. Results are presented in Table 3 for ResNet and Table 6 in Appendix D for ViT.

The Hyperprior model illustrates classic gradient masking behavior. It performs well against attacks relying on gradients computed through the defense, but fails under black-box attacks or gray-box attacks with techniques like U-Net BPDA or ST BPDA. In those settings, its robust accuracy collapses.

In contrast, CRDR HR shows strong robustness across attacks. WB PGD, using true gradients, is the most effective or close second, indicating CRDR isn't just masking gradients. However, at high perturbations ($\epsilon = \frac{16}{255}$), U-Net BPDA sometimes outperforms WB PGD, suggesting that out-of-distribution inputs reduce true gradient effectiveness and increase gradient masking. In such cases, surrogate-based gradients like U-Net BPDA yield stronger attacks.

### 4.4 Comparison to Diffusion-Based Defenses

As noted earlier, diffusion models can purify adversarial noise; however, they are computationally expensive. For example, Lee and Kim [23] takes over 60 minutes to process 100 images on an RTX 3090, while CRDR needs only 1.5 seconds. With the ResNet classifier taking 0.5 seconds, CRDR increases inference time $4\times$, compared to over $7200\times$ for the diffusion method. Nevertheless, we compare CRDR HR to diffusion models due to their high realism. As shown in Table 4, VAE-based models like CRDR HR offer comparable robustness at much lower cost, making them a more practical choice for attack evaluations while maintaining strong defensive performance.

We also conduct our own small-scale test (in terms of the number of data samples) to assess the robustness of the diffusion models. Due to space, the results are in the Appendix, but show that a PGD 8/255 attack can get an ASR around 72%, giving a model accuracy around 20% for the diffusion model by Lee and Kim [23].

### 4.5 Highlight of Extra Results in the Appendix

Due to space constraints, additional results are provided in the appendix for interested readers.

| Defense Method | Standard | PGD 4/255 |
|---|---|---|
| Engstrom et al. [52] | 62.42 | 33.20 |
| Wong et al. [53] | 53.83 | 28.04 |
| Nie et al. [19] | 75.48 | 38.71 |
| Lee and Kim [23] | 66.21 | 42.15 |
| CRDR | 61.72 | 35.88 |

Table 4: Robust accuracy under PGD 4/255 attack running for ten iterations for diffusion-based defenses and the CRDR compression-based defense. The robust accuracy numbers of the diffusion-based models are taken from [23].

The section titled "Attacking Adversarial Purification" demonstrates that while it is computationally expensive, diffusion models can be attacked.

Instead of evaluating against standard (adversarially weak) classifiers, one can use pretrained robust classifiers to assess whether realism offers additional benefits in already robust settings. We ran the compression-based defense on the top 11 models from RobustBench [54, 15, 55–58]. Our RobustBench results can be seen in Table 11 in Appendix G.3. The results show that applying CRDR with high quality and realism does not improve performance—in fact, it consistently reduces accuracy across all tested robust models.

Iterative compression as a defense has shown promising results [2], but its robustness largely stems from gradient masking. When attacking the defense using fewer iterations, robust accuracy drops significantly, revealing the underlying vulnerability once gradient masking is circumvented. See Appendix G.4 for detailed results.

Interestingly, under a white-box PGD attack, the structure of the adversarial perturbation tends to follow the image structure, see Appendix G.5. In particular, the perturbation primarily affects object edges and struggles to inject noise uniformly across the image. Realism amplifies the model's tendency to hallucinate details—realistic models generate finer, more structured outputs. These hallucinated details may change when attacking such models, but the resulting images still appear realistic to the human eye.

## 5 Discussion and Future Work

In this work, we systematically evaluated attacks against compression-based adversarial defenses, identifying what makes specific compression models difficult to break. Our findings reveal a critical challenge for attackers: high realism in reconstructed images significantly increases attack difficulty. While most compression-based defenses fail under adaptive attacks, models with high realism consistently demonstrate greater resistance.

Through rigorous experimentation, we have shown that realism, not gradient masking or other obfuscation techniques, is the primary obstacle for attackers. When compression models produce realistic reconstructions, they maintain distributional alignment with natural images while discarding adversarial perturbations, creating a fundamental asymmetry that favors the defender. Attackers must find perturbations that survive the compression process and remain effective after high-quality reconstruction, a significantly more challenging task.

Future work should focus on developing more effective attacks against high-realism compression models. This includes designing loss functions that better capture realistic reconstruction quality and crafting attacks that target preserved semantic features rather than merely pixel-level noise. Efficiently attacking diffusion-based compression methods, which naturally achieve high realism, also remains a key challenge for thorough security evaluation. The concept of *perfect realism*, where reconstructions are indistinguishable from natural data and lie on the data manifold, offers an ideal defense by projecting inputs onto the natural image distribution and eliminating adversarial perturbations. Should attacks on these high-realism models, especially diffusion-based approaches, continue to fail, our findings would mark a promising step toward genuinely robust defenses.

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

# A    Limitations

The primary limitation of our study is the exclusion of diffusion-based compression models, known to achieve the highest levels of realism. This omission was due to computational constraints but represents an important direction for future research. Additionally, we did not run experiments across multiple random seeds to quantify variance. Although the inherent stochasticity of PGD introduces some variability, incorporating multiple seeds would significantly increase computational costs.

# B    Extended Related Work and Background

## B.1    Adversarial Robustness

Shortly after the success of AlexNet [8], it was found that neural networks are very susceptible to *adversarial attacks* [9, 10]. Here, an adversary makes, usually small and imperceptible, modifications (perturbations) to, for instance, an image such that a model mislabels it.

**Attacks**    Many attacks have been developed over the years with various benefits and drawbacks. Some of the most noteworthy are FGSM (Fast Gradient Sign Method) [10], iterated FGSM or iFGSM [11], CW (Carlini & Wagner) [12], and PGD (Projected Gradient Decent) [13]. Additional attacks that have often been used are APGD [59], DeepFool [60], and ZOO [61]. We refer the reader to the original papers for specific details, but we include the necessary details to understand this paper in Appendix C. The main thing to know is that PGD has a perturbation budget $\epsilon$ and number of iteration it can perform $n$. PGD then does projected gradient decent for $n$ iterations to find an adversarial example that is at most $\epsilon$ distance in $l_\infty$ from the original image.

The above attacks fall into three broad categories: FGSM, iFGSM, PGD, and APGD are $l_\infty$-bounded attacks, CW and DeepFool are $l_2$-bounded attacks, and ZOO is a gradient-free attack using the predicted confidence scores for each class. PGD can also be implemented as an $l_2$-bounded attack.

When the defense includes randomness, a common attack augmentation is EoT (Expectation over Transformation), in which gradients are averaged over multiple forward and backward passes [14, 7]. If the defense causes the gradients to be infeasible to compute, Athalye et al. [14] suggested Backward Pass Differentiable Approximation (BPDA), where the defense is used during the forward pass, but a differentiable approximation is used during the backward pass. The simplest is to use the identity function, while more advanced methods might train a differentiable surrogate model $g'$ that emulates the defense $g$; i.e., $g(x) \approx g'(x)$.

Moosavi-Dezfooli et al. [62] show in their work "Universal adversarial perturbations" that adversarial perturbations transfer to other data samples and models; Szegedy et al. [9] already hinted at this years before. Several later studies have explored this [63–65], attempted to explain the effect [66],

and shown the effect also exists for language models [67]. The transfer effect allows for another frequently used technique of black box attacks, where perturbations are generated for one model and applied to a black box model [4]. Another branch of black box attacks looks only at the prediction labels of a model to generate attacks [68].

However, as highlighted by Sheatsley et al. [6], these are only a small set of all the attacks that could be considered. The list of possible attacks is heavily dependent on the threat model used [4, 6], and if one relaxes the often-used imperceptible assumption, then there exists a wide range of semantic-preserving attacks [69–71], and natural adversarial examples [72, 73].

Lastly, often the best attack results are found using *adaptive attacks* [4–7]. Here, the attacks, for instance, the optimization objective, are adjusted to the defense. It is thus vital to consider adaptive attacks when a defense is resistant to standard gradient-based attacks. This work focuses on adaptive attacks with PGD as the underlying optimization method.

**Defenses**   Given the prevalence of adversarial attacks, many researchers have explored how to defend against them. We broadly view this in three groups: Architecture improvements [15–17], adversarial training [9, 10, 13, 18, 17], and adversarial purification [19–21, 2].

The architecture branch focuses on making the models more robust by design, for instance, by taking inspiration from biology to mimic the human eye when training a ResNet model [17].

Adversarial training, as formalized by Madry et al. [13], is perhaps the most straightforward method and consists of showing the model adversarial examples during training to ensure it classifies these correctly. This method has yielded positive results [9, 13], but models with adversarial training can be broken [7]. They also experience a drop in clean accuracy (accuracy on images without adversarial perturbations) [13, 4], and the robustness may not carry over to other attacks [7].

The idea of adversarial purification is to remove the adversarial noise in images before passing the cleaned images to pretrained classification models. It is motivated by the work of Ilyas et al. [22], hinting that adversarial examples perturb brittle features in the model. These methods *should work independently* of any robustness applied to the classifier through adversarial training or robust architectures. Two noteworthy works using diffusion-based models to remove the adversarial noise are Nie et al. [19], Lee and Kim [23].

In the realm of adversarial purification, diffusion models have been proposed as a way to remove adversarial noise from input images [19, 23]. These approaches are conceptually similar to compression-based defenses with realism: both aim to project adversarial examples back onto the manifold of natural images to restore classifier performance. However, prior work on diffusion-based purification has not explicitly investigated the role of realism as a contributing factor to robustness.

One major drawback of diffusion-based defenses is their computational cost. Diffusion models are typically an order of magnitude more expensive than VAE- or GAN-based learned compression models, making them impractical for many real-world settings [24, 25]. This high cost limits their deployment and complicates the evaluation of robustness: mounting effective adversarial attacks against diffusion models becomes significantly harder due to the computational overhead. However, simply making gradients difficult to compute does not equate to genuine robustness. A well-designed adaptive attacker, given sufficient resources, should still be able to circumvent such defenses [4, 23].

Notably, the purification method proposed by Nie et al. [19] was later defeated by Lee and Kim [23]. However, the latter only evaluated their improved defense under the attack that broke the former. They did not develop new adaptive attacks tailored to their defense—a strategy that past research suggests would likely reduce the effectiveness of the defense. The history of adversarial robustness research shows that defenses that are not tested under strong, tailored attacks often overstate their robustness [14, 4, 5].

**RobustBench**   The considerable interest in adversarial examples has given rise to benchmarks and leaderboards, one of which is RobustBench [54]. This provides a leaderboard of models that are supposed to be robust to adversarial examples. However, note that Fort and Lakshminarayanan [17] claimed their model beat the RobustBench leaderboard, but, using adaptive attacks, Zhang et al. [7] showed that the model is still very vulnerable to adversarial examples.

We will use RobustBench as a source of robust models, and thus enable us to test if compression-based purification would help as claimed in [2]. For this, we take top-performing models from the leaderboard with open-sourced model weights and test whether the compression-based purification defenses improve the models' robustness.

## B.2 Realism

Image compression algorithms are traditionally evaluated using *distortion metrics* such as PSNR or SSIM, which measure the distance between a restored image $\hat{x}$ and its reference $x$. Formally, distortion is defined as:

$$\mathcal{D} := \mathbb{E}_{(x,\hat{x})\sim(p_X,p_{\hat{X}})}[\Delta(x,\hat{x})], \tag{4}$$

where $\Delta(\cdot,\cdot)$ is a pointwise distortion measure (e.g., $\ell_2$ distance), and $p_X$, $p_{\hat{X}}$ denote the distributions of ground truth and reconstructed images. Distortion is considered a *full-reference* metric, requiring access to the original image $x$ to compare it to the reconstructed version $\hat{x}$.

However, metrics such as PSNR, MS-SSIM [26], or LPIPS [27] correlate poorly with human perception, and directly quantifying perceptual distortion remains a challenging problem. Instead of a perceptual distortion metric, one can also measure both distortion and realism and optimize the compression model for both. Realism can be formally defined as:

$$\mathcal{R} := -d(p_{\hat{X}},p_X), \tag{5}$$

where $d(\cdot,\cdot)$ is a divergence measure, such as Kullback-Leibler. Unlike distortion, realism is a *no-reference metric*, requiring only the generated image distribution to match that of natural images. Although measuring realism remains challenging [28], a widely used proxy is the Fréchet Inception Distance (FID) [29], which compares the distributions extracted by a pretrained Inception network. FID has gained popularity for capturing both fidelity and diversity in generated samples.

Compression models are typically trained with the following loss function:

$$\mathcal{L} = \mathcal{L}_{\text{RATE}} + \lambda\mathcal{D} - \beta\mathcal{R}, \tag{6}$$

where $\mathcal{L}_{\text{RATE}}$ represents the estimated rate, or in other words, the number of bits required to represent the image after it has been compressed, $\lambda$ controls the level of distortion, and $\beta$ the level of realism. Since the information is not transmitted through an explicit information bottleneck in our approach, the rate term $\mathcal{L}_{\text{RATE}}$ does not play a critical role in this context.

Blau and Michaeli [74] prove that there exists an inherent tradeoff between distortion $\mathcal{D}$ and realism $\mathcal{R}$. Specifically, reducing one inevitably increases the other, regardless of the choice of distortion or divergence. This theoretical result underpins the empirical observation that GAN-based methods, which maximize $\mathcal{R}$ through adversarial training, tend to increase $\mathcal{D}$ while producing perceptually convincing outputs.

Our work builds on this foundation, exploring the intersection of realistic compression and adversarial robustness. We extend the existing literature by providing experimental evidence that realism, rather than distortion minimization, makes image-compression models (partially) robust against adversarial examples.

## B.3 Compression Models

End-to-End Optimized Image Compression [31] jointly learns analysis and synthesis transforms along with an entropy model over quantized latents, directly optimizing rate–distortion performance in a VAE-style framework. The Hyperprior model [45] extends this approach by introducing a secondary network that predicts spatially adaptive Gaussian scales, better capturing local image statistics. HiFiC [33] enhances learned compression with GANs to produce realistic reconstructions. By refining network design and training with perceptual losses, it generates outputs that closely resemble natural images.

ELIC [32] introduced uneven channel grouping and deeper autoregressive context networks for more accurate entropy modeling, yielding faster convergence and lower bitrates. PO-ELIC [34] then augmented this foundation with adversarial fine-tuning and perceptual losses to enrich texture realism at low bitrates.

More recent work has shifted toward models that offer explicit, user-controllable trade-offs between rate, distortion, and realism within a single network. MRIC [37] introduced a conditional generator that, at a fixed bit rate, lets users interpolate between low-distortion and high-realism reconstructions by tuning a realism flag $\beta$, thereby explicitly navigating the distortion–realism trade-off within a single model. CRDR [38] built on this by adding a discrete quality-level input $q$ alongside $\beta$, and embedding interpolation channel attention layers in both encoder and generator to yield true variable-rate compression, allowing joint control over bitrate, distortion, and realism with one network.

Diffusion models are great at generating images, which is also taken advantage of in image compression. Models [75–79] that focus on extra low bitrate or extra high realism without worrying about compute use diffusion. These methods are orders of magnitude more expensive than VAE-based methods, but achieve great realism.

Reducing computational complexity has also been a focus in compression with the line of work like Cool-Chic [80] and C3 [81]. These methods overfit a latent, auto-regressive latent model and decoder to a single image, and send all three over the channel. High realism can be achieved at this level of complexity [82], but as these models are INRs, computing a gradient through the compression model is not straightforward, and encoding is computationally expensive.

### B.4 Compression as adversarial defense

Using compression as a defense for neural networks is not a new idea. It has been explored for almost a decade [1, 39–42] where some authors also explored using iterated compression and decompression cycles [42, 2]. And more recently, using video compression to defend video classifiers [83] and model quantization to defend general models [84].

Two key works in this area are by Guo et al. [1] and Shin and Song [3]; the former argued that JPEG compression as a preprocessing step is a very effective adversarial defense while the latter showed that, by making JPEG differentiable, this defense could be bypassed entirely–highlighting the necessity of properly evaluating a defense.

## C    Details on Adversarial Attacks

PGD is a very strong attack, and most defenses in a white-box setting can be broken by it when using the right objective.[1]

For the rest of this section, let $f$ be a neural network, $L$ a loss function, $x$ an input with corresponding output $y$, and $\nabla_x f(x)$ the gradient of $L(f(x), y)$ with respect to $x$.

FGSM computes the gradients of the loss function with respect to the image's pixel values, takes the sign of the gradients, and multiplies by $\epsilon$, where $\epsilon$ is a small number, usually $\frac{4}{255}$ for ImageNet images and $\frac{8}{255}$ for CIFAR-10 images. The attack can be written as:

$$FGSM(f, x) = x + \epsilon \cdot \text{sign}(\nabla_x f(x)).$$

iFGSM uses two extra parameters $\alpha < \epsilon$ and $n$. iFGSM then takes $n$ FGSM steps, by default $n = 10$, of size $\alpha$ ensuring the final perturbation is within the $l_\infty$ ball of size $\epsilon$.

$$iFGSM(f, x) = x_n$$
$$x_0 = x$$
$$x_{i+1} = Clip_{x,\epsilon}\left(x_i + \epsilon \cdot \text{sign}(\nabla_{x_i} f(x_i))\right)$$

where $Clip_{x,\epsilon}$ clips $x_i$ to be within the $l_\infty$ ball of radius $\epsilon$ of $x$.

PGD works almost entirely like iFGSM; however, there are two key changes. 1) It randomly initializes $x_0$ in the $\epsilon$ $l_\infty$-ball. 2) With the stochasticity from 1), it includes the option for restarts, and thus running the optimization again.

---

[1]Personal communication with Nicholas Carlini with further evidence in [7].

| Strength | Defense | Standard | BB PGD | WB PGD | ST BPDA | U-Net BPDA | ACM | ARA |
|---|---|---|---|---|---|---|---|---|
| 4/255 | Hyperprior | 78.73 | 48.76 | 78.84 | 10.94 | **0.98** | 78.82 | — |
| | HiFiC | 61.53 | 59.52 | **11.83** | 44.65 | 24.04 | 59.20 | — |
| | MRIC LR | 52.80 | 51.04 | **26.68** | — | — | — | **26.68** |
| | MRIC HR | 63.06 | 59.28 | **39.00** | — | — | — | 39.60 |
| | CRDR LR | 46.02 | 44.92 | **16.30** | 39.36 | 28.96 | 41.28 | **16.30** |
| | CRDR HR | 61.72 | 59.80 | 35.88 | 56.36 | 47.62 | 55.67 | **34.50** |
| | JPEG | 70.30 | 65.68 | 8.72 | 18.70 | **5.19** | 68.67 | — |
| | ELIC | 60.03 | 58.79 | **16.43** | 40.00 | 17.98 | 54.48 | — |
| 8/255 | Hyperprior | 78.73 | 27.16 | 78.72 | 3.18 | **0.02** | 78.67 | — |
| | HiFiC | 61.53 | 56.84 | **6.65** | 32.19 | 7.98 | 55.37 | — |
| | MRIC LR | 52.80 | 49.44 | **12.20** | — | — | — | **12.20** |
| | MRIC HR | 63.06 | 57.38 | 21.98 | — | — | — | **21.10** |
| | CRDR LR | 46.02 | 44.14 | **8.60** | 33.58 | 12.12 | 36.96 | **8.60** |
| | CRDR HR | 61.72 | 57.36 | 23.92 | 50.77 | 26.64 | 49.35 | **21.18** |
| | JPEG | 70.30 | 60.76 | 5.19 | 10.01 | **0.24** | 64.75 | — |
| | ELIC | 60.03 | 57.59 | 9.01 | 27.24 | **4.98** | 51.63 | — |
| 16/255 | Hyperprior | 78.73 | 6.99 | 76.94 | 1.96 | **0.00** | 76.82 | — |
| | HiFiC | 61.53 | 51.67 | 3.08 | 25.53 | **1.80** | 45.17 | — |
| | MRIC LR | 52.80 | 49.40 | **2.90** | — | — | — | **2.90** |
| | MRIC HR | 63.06 | 52.90 | 7.26 | — | — | — | **6.40** |
| | CRDR LR | 46.02 | 41.40 | **4.10** | 30.38 | 2.96 | 27.82 | **4.10** |
| | CRDR HR | 61.72 | 52.70 | 13.93 | 47.10 | **7.98** | 36.68 | 11.42 |
| | JPEG | 70.30 | 49.48 | 2.86 | 6.58 | **0.01** | 52.99 | — |
| | ELIC | 60.03 | 54.68 | 4.10 | 20.62 | **0.34** | 36.47 | — |

Table 5: Extended version of Table 3 with the 16/255 results. Results for ResNet50. "—" denotes values not implemented for MRIC or evaluations invalid without realism control. Models with strong gradient masking, like Hyperprior, are vulnerable to adaptive attacks. Realism does not increase gradient obfuscation, so both high-realism models retain accuracy with minor drops under adaptive attacks.

## D Extended adaptive attacks

We show in Table 5 complete results for the adaptive attacks applied to the ResNet model.

We show in Table 6 results for the adaptive attacks applied to the ViT model.

## E Hyperparameters

This section includes information about the hyperparameters used during our experiments. Unless stated otherwise (e.g, number of steps or epsilon as columns/rows of the table), we use the following in Tables 7 and 8.

For the Adaptive Realism attack, the realism parameter was chosen amongst the beta values specified in Table 8. The results of this experiment can be found in Table 9.

We use the 50000 images from the ImageNet validation set for most of our results. Exceptions are the U-Net BPDA in Table 3 and Table 6, the results in table Table 2 and the results of the ARA ablation in Table 9

## F Training U-Nets for BPDA

The U-Net used to approximate the compression and decompression step in the BPDA attack follows a standard U-Net architecture [50]. We use two downsampling layers, which transform the input images from a resolution of (3,224,224) to an intermediate representation of (256,56,56). For each experiment, we train the U-Net on the whole dataset used in the experiment. The U-Net is trained for 20 epochs using the L1-loss between the input image (original image with or without adversarial

| STRENGTH | DEFENSE | STANDARD | BB PGD | WB PGD | U-NET BPDA | ST BPDA | ACM | ARA |
|---|---|---|---|---|---|---|---|---|
| 4/255 | HYPERPRIOR | 80.38 | 7.47 | 80.51 | 1.72 | **0.20** | 80.46 | |
| | HIFIC | 70.76 | 62.12 | **10.06** | 39.78 | 22.56 | 69.32 | |
| | CRDR LR | 52.24 | 49.24 | **14.44** | 24.82 | 33.38 | 48.64 | **14.44** |
| | CRDR HR | 68.82 | 62.30 | 28.28 | 53.22 | 41.76 | 63.90 | **27.28** |
| | JPEG | 74.80 | 36.86 | 2.70 | 5.54 | **0.88** | 73.92 | |
| | ELIC | 68.24 | 60.44 | 16.21 | 27.36 | **14.12** | 63.16 | |
| 8/255 | HYPERPRIOR | 80.38 | 0.61 | 80.25 | 0.16 | **0.00** | 80.38 | |
| | HIFIC | 70.76 | 56.30 | **2.64** | 24.68 | 4.00 | 67.12 | |
| | CRDR LR | 52.24 | 46.14 | **6.98** | 13.10 | 21.12 | 43.16 | **6.98** |
| | CRDR HR | 68.82 | 57.43 | 16.68 | 39.60 | 14.90 | 60.44 | **13.86** |
| | JPEG | 74.80 | 15.34 | 0.94 | 1.14 | **0.04** | 71.50 | |
| | ELIC | 68.24 | 54.82 | 6.20 | 15.22 | **2.38** | 59.99 | |
| 16/255 | HYPERPRIOR | 80.38 | 0.01 | 78.65 | 0.12 | **0.00** | 78.48 | |
| | HIFIC | 70.76 | 46.24 | 0.54 | 14.32 | **0.16** | 59.56 | |
| | CRDR LR | 52.24 | 40.60 | **2.16** | 8.74 | 14.08 | 32.32 | **2.16** |
| | CRDR HR | 68.82 | 49.48 | 9.31 | 28.38 | **2.46** | 50.62 | 6.34 |
| | JPEG | 74.80 | 1.66 | 0.28 | 0.54 | **0.00** | 62.80 | |
| | ELIC | 68.24 | 43.70 | 1.00 | 8.52 | **0.10** | 45.94 | |

Table 6: Results for ViT

| CRDR LR | QUALITY=0 | $\beta = 0$ |
|---|---|---|
| CRDR HR | QUALITY=0 | $\beta = 5.12$ |
| MRIC LR | WEIGHTS=128 | $\beta = 0$ |
| MRIC HR | WEIGHTS=128 | $\beta = 2.56$ |
| HYPERPRIOR | QUALITY=8 | |
| HIFIC | WEIGHTS=LOW | |
| ELIC | WEIGHTS=0016 | |
| JPEG | QUALITY=25 | |

Table 7: Hyperparameters used for the compression models. Weights indicate the designation of the pretrained weights used, quality indicates the quality parameter for compressions with variable quality, and beta indicates the realism parameter for compressions with variable realism.

noise) and the target image (reconstruction of the input image) using an Adam optimizer with a learning rate of 0.001 and a learning rate schedule. The learning rate scheduler is StepLR from PyTorch, configured with a step size of 5 and a gamma of 0.1. During an epoch, each batch is used 4 times, 3 passes with additional noise added to the images to simulate adversarial noise, and one clean pass with the original images.

# G Extended results

## G.1 Attacking Adversarial Purification

We ran a limited attack on a diffusion-based adversarial purification defense [23]. Since running (and attacking) this defense is more computationally expensive compared to the compression models used in the other results shown in this paper, we focused on a smaller part of the ImageNet dataset.

We ran a U-Net BPDA attack on 100 images. Table 10 shows the accuracy of an attack with epsilon 8/255. The attacked accuracy of 69% shows that the U-Net attack was not able to produce practical gradients, as the accuracy is higher than the base accuracy. The attack proposed by [23] showed an accuracy of 42.15% with an even lower epsilon of 4/255. In addition, we apply a very aggressive

| | | | | | | | $\beta$ | | | |
|---|---|---|---|---|---|---|---|---|---|---|
| PGD | $\epsilon = 8/255$ | $\alpha = \epsilon/4$ | | STEPS$= 10$ | | | RANDOM START=TRUE | | | |
| ST BDPA | $\epsilon = 8/255$ | $\alpha = \epsilon/4$ | | STEPS$= 10$ | | | | | | |
| U-NET BDPA | $\epsilon = 8/255$ | $\alpha = \epsilon/4$ | | TRAINING EPOCHS$= 20$ | | ATTACK STEPS $= 100$ | | | | |
| ACM | $\epsilon = 8/255$ | LOSS = MSE | | STEPS$= 20$ | | | | | | |
| ARA | ATTACK=PGD | $\beta \in \{0.0, 0.16, 0.32, 0.64, 1.28, 2.56, 5.12\}$ | | | | | | | | |

Table 8: Hyperparameters used for the different attacks. Epsilon is the maximum perturbation allowed for an adversarial image, alpha controls the maximum perturbation added per step.

| MODEL | DEFENSE | STANDARD | $\beta$ | | | | | | |
|---|---|---|---|---|---|---|---|---|---|
| | | | 0 | 0.16 | 0.32 | 0.64 | 1.28 | 2.56 | 5.12 |
| RESNET50 | CRDR LR | 46.02 | **8.60** | 14.98 | 17.30 | 21.14 | 23.08 | 26.72 | 29.84 |
| | CRDR HR | 61.72 | 28.82 | 22.12 | 21.70 | **21.18** | 21.60 | 22.90 | 24.32 |
| | MRIC LR | 52.80 | **12.20** | 22.60 | 24.60 | 28.10 | 30.10 | 31.40 | — |
| | MRIC HR | 63.06 | 31.90 | 25.20 | 22.40 | 21.80 | **21.10** | 21.98 | — |
| VIT | CRDR LR | 52.24 | **6.98** | 11.06 | 12.12 | 14.68 | 16.10 | 19.46 | 24.50 |
| | CRDR HR | 68.82 | 19.12 | 14.82 | **13.86** | 14.20 | 14.64 | 15.76 | 17.02 |

Table 9: Comparison of different realism values used in the Adaptive Realism Attack at $\epsilon = \frac{8}{255}$. CRDR LR uses $\beta = 0$ in the defense, CRDR HR uses $\beta = 5.12$. The bold values represent the strongest atttack an were used in Table 3 and Table 6.

attack with an epsilon of 64/255. However, Table 10 shows that the results are comparable to adding random noise to the image, showing that the gradients carry no usable signal.

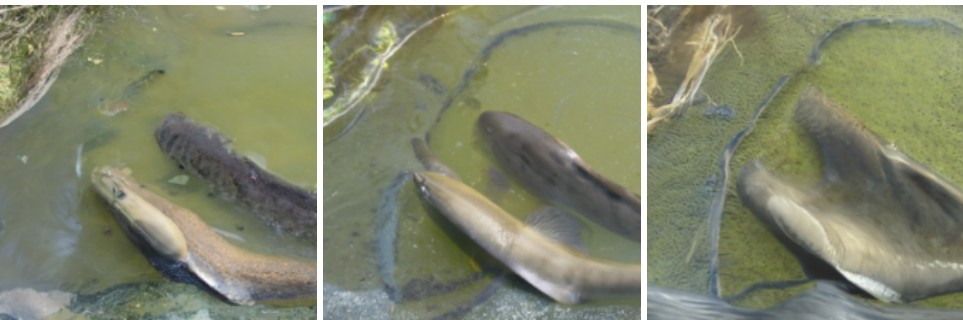

Figure 4: Three different diffusion outputs for the same input image. These showcase the large differences the diffusion models can introduce and thus what the adversarial noise must be robust to.

We assume that the U-net failed to capture the large variance shown by the diffusion model. Figure 4 shows the large visible variance in diffusion output images. Since the training, especially the creation of a dataset of diffusion input-output image pairs, was computationally expensive, we did no further experiments with the U-net attack–we utilized approximately 2,000 additional GPU hours for this experiment.

We also ran the PGD + EOT attack from [23] for epsilon 8/255 and 100 PGD iterations. The weaker defnse we attacked used just 9 diffusion steps. This allows us to compute the gradient through the entire defense. To better evaluate our results, we compute the accuracy of the full defense + classifier after every PGD iteration. We started the attack with 64 correctly classified images. Figure 5 shows the decrease in accuracy over 100 PGD iterations. The non-deterministic nature of the diffusion model leads to a certain variance in the accuracy at higher iterations. We therefore report the average accuracy of 28.1% over iterations 50 to 100 and the minimal accuracy of 21.9%. The bottom graph in Figure 5 shows the percentage of images that have never been missclassified up to the current PGD iteration. Evaluating this metric provides an estimate of worst-case adversarial robustness, which is 14.1% after 100 iterations.

| ATTACK | EPSILON | ACCURACY | |
|--------|---------|----------|---------|
| | | CLEAN | ATTACKED |
| U-NET | 8/255 | 68.0 | 69.0 |
| U-NET | 64/255 | 63.8 | 12.0 |
| NOISE | 64/255 | 61.3 | 27.0 |
| PGD + EOT | 4/255 | 70.7 | 42.15 |

Table 10: Accuracy of the U-net attack on adversarial purification. The PGD + EOT are from [23].

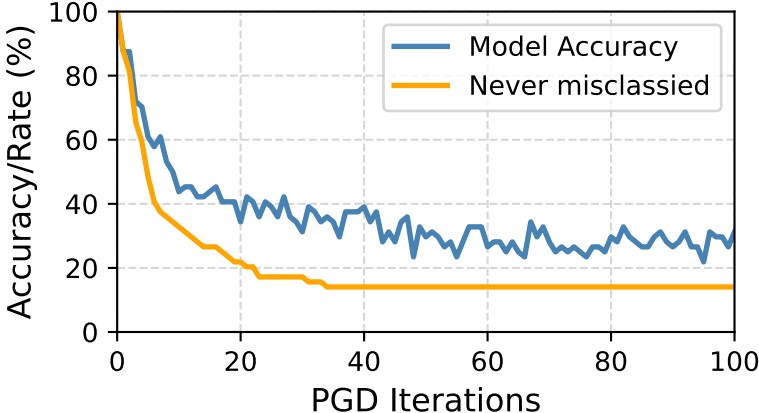

Figure 5: Accuracy of the adversarial purification defense for 100 iterations of PGD. The blue curve shows the accuracy for every iteration, and the orange curve shows the fraction of images that have never been misclassified up to that iteration.

The numbers reported above are for a subset with 100% clean accuracy (i.e., they correspond to 100%−ASR). Thus, after accounting for the model accuracy of 70.7%, the average accuracy is 19.9%, placing it only slightly higher in accuracy than CRDR with high realism (cf. Table 2).

The results of this second experiment show that evaluating a diffusion-based defense not only requires significant computational effort but is also inherently more challenging due to the variance in model output, as incorrect predictions can occur for otherwise correctly classified images. This makes direct comparisons of the attack success rate (ASR) to other defenses more challenging.

For the second attack on diffusion-based purification models that was successful, we used two A100s for two days. Thus, scaling up the attacks will be difficult as this puts the processing rate at 16 images *per day* for an A100. Faster diffusion models, such as the model by Lei et al. [85], might make larger-scale experiments feasible; however, the authors do not provide trained model weights.

### G.2 Distortion vs Realism

As shown in Figure 6, robust accuracy increases monotonically with realism. This trend does not hold for distortion: there exists an optimal level of distortion that balances preserving informative content from the original image while removing adversarial perturbations. Interestingly, the optimal distortion level shifts higher as realism increases. This can be attributed to artifacts introduced by compression—excessive artifacts can act as adversarial perturbations themselves. By incorporating realism that mitigates such artifacts, stronger compression-based defenses become possible. For many models, performance gains from increased realism have not yet saturated, although these models were not originally designed to operate at higher values of the $\beta$ parameter. While prior work has established that a certain distortion (or quality) level yields optimal adversarial robustness with compression, our work is the first to systematically investigate the role of realism. We demonstrate that defenses lacking realism are significantly easier to attack.

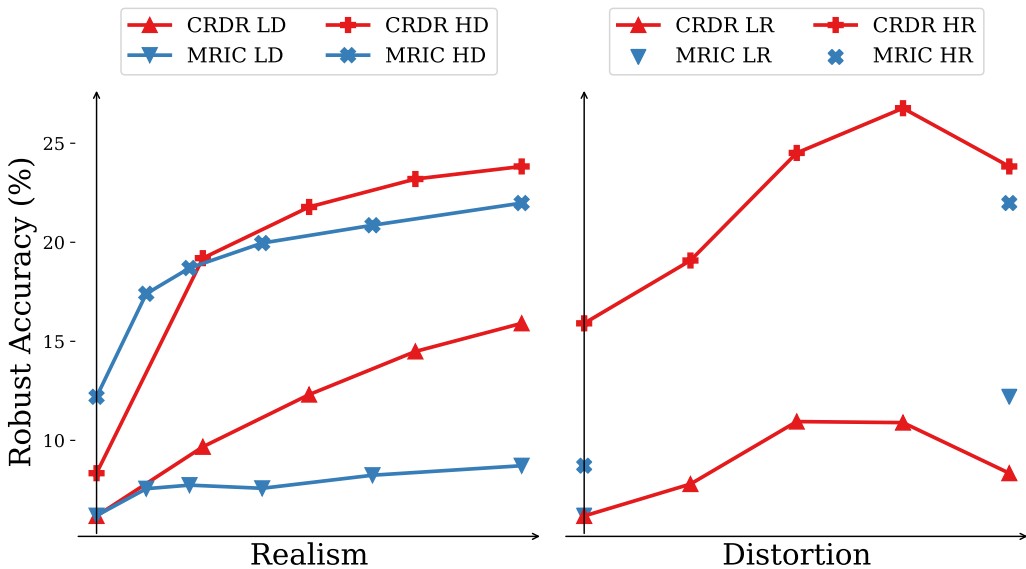

Figure 6: Impact of realism and distortion on robust accuracy for CRDR and MRIC. Realism and distortion are measured by the training parameters $\lambda$ and $\beta$, respectively, and normalized to their minimum and maximum values. Note that for MRIC, pretrained weights are only available at two distortion levels. Higher realism in reconstructed images improves robustness against adversarial attacks. However, excessive distortion can degrade accuracy, suggesting the existence of an optimal distortion level that balances detail preservation and defense effectiveness.

### G.3 RobustBench

Instead of evaluating against standard (adversarially weak) classifiers, one can use pretrained robust classifiers to assess whether realism offers additional benefits in already robust settings. We ran the compression-based defense on the top 11 models from RobustBench [15, 55–58]. The results in Table 11 show that applying CRDR with high quality and high realism does not improve performance—in fact, it consistently reduces accuracy across all tested robust models. While CRDR benefits standard models by projecting inputs back onto the natural image manifold, we see two reasons for these results. 1) The information loss and subtle degradations introduced by compression can harm standard and robust accuracy when applied to already robust models, as the compression may discard the specific features these robust models have learned to utilize for classification, explaining why performance decreases rather than increases. 2) The robustified models have been overfitted to ImageNet images ([86, 87]) without *any* noise and images with *exactly* the type of noise PGD produces. We leave it for future work to resolve which, if any, of these explanations are correct.

### G.4 Iterative Defenses

It was claimed in prior work that applying compression iteratively strengthens adversarial defenses [2]. However, we show that this effect is primarily due to gradient masking rather than true robustness (cf. Table 12 in Appendix G.4). When attacking an iterative defense by approximating gradients using fewer defense iterations, the gradients remain informative, and accuracy can be reduced to levels comparable to using a single defense iteration.

As shown in Figure 7, the most effective defense configuration for CRDR involves multiple defense iterations with low compression quality and high realism. Interestingly, the optimal attack against this configuration uses fewer iterations to approximate the gradients. In contrast, attacks that match the defense in the number of iterations perform worse. This further indicates that CRDR still exhibits some gradient masking, which standard PGD attacks fail to overcome fully (cf. Section 4.3).

| | STANDARD | | 4/255 | | 8/255 | | 16/255 | |
|---|---|---|---|---|---|---|---|---|
| MODEL | BASE | CRDR | BASE | CRDR | BASE | CRDR | BASE | CRDR |
| AMINI CONVNEXT-L [56] | **78.58** | 76.04 | **61.98** | 57.08 | **43.98** | 36.20 | **18.30** | 11.84 |
| AMINI SWIN-L [56] | **78.98** | 77.10 | **65.12** | 60.18 | **49.42** | 41.66 | **25.06** | 17.28 |
| BAI NUTS [55] | **81.48** | 80.12 | **70.66** | 66.76 | **51.10** | 46.04 | **14.28** | 11.62 |
| LIU CONVNEXT-B [58] | **77.16** | 74.48 | **58.40** | 53.34 | **39.44** | 33.06 | **18.34** | 13.48 |
| LIU CONVNEXT-L [58] | **78.62** | 76.18 | **60.48** | 55.26 | **41.44** | 34.32 | **19.80** | 14.24 |
| LIU SWIN-B [58] | **76.78** | 74.62 | **59.80** | 53.80 | **41.08** | 34.60 | **19.80** | 13.84 |
| LIU SWIN-L [58] | **79.00** | 77.12 | **61.94** | 57.04 | **43.04** | 36.76 | **22.24** | 16.00 |
| SINGH CONVNEXT-B [15] | **75.94** | 73.52 | **58.00** | 52.18 | **38.84** | 32.62 | **16.18** | 12.08 |
| SINGH CONVNEXT-L [15] | **77.66** | 75.16 | **60.32** | 54.84 | **41.86** | 35.24 | **19.12** | 15.02 |
| XU SWIN-B [57] | **77.26** | 74.90 | **58.58** | 53.62 | **39.18** | 32.88 | **16.32** | 12.40 |
| XU SWIN-L [57] | **79.40** | 77.30 | **62.32** | 57.00 | **42.20** | 35.86 | **19.06** | 14.72 |

Table 11: Accuracy (%) for RobustBench models attacked by PGD and then defended using iterative CRDR in a white-box setting. Bold highlights better performance between Base and CRDR for each pair. We take the 11 top-performing models from https://github.com/RobustBench/robustbench under ImageNet.

| ITERATIONS | | | |
|---|---|---|---|
| DEFENCE | ATTACK | STANDARD | PGD |
| 50 | 50 | 69.92 | 67.92 |
| 50 | 25 | 69.92 | 65.74 |
| 50 | 10 | 69.92 | 58.76 |
| 50 | 5 | 69.92 | 43.56 |
| 50 | 1 | 69.92 | 7.10 |
| 10 | 10 | 69.94 | 58.94 |
| 10 | 5 | 69.94 | 43.20 |
| 10 | 1 | 69.94 | 7.02 |
| 1 | 1 | 71.54 | 5.50 |
| 50 | 25 | 69.92 | 65.74 |
| 25 | 25 | 69.92 | 65.72 |
| 10 | 25 | 69.94 | 65.90 |
| 5 | 25 | 69.98 | 66.04 |
| 1 | 25 | 71.54 | 68.14 |
| 0 | 25 | 80.64 | 78.94 |

Table 12: Accuracy (%) for iterative JPEG defenses as in [2]. Performing the attack (PGD with epsilon 8/255) on a defense with just one iteration defeats the iterative version. The robust accuracy seems connected to the number of iterations in the attack, not the defense.

### G.5  Structure of the adversarial noise

As shown in Figure 8, CRDR structures the adversarial noise. The image cannot be altered unstructured; the compression model ensures that the perturbation follows the image's inherent structure. When attacking CRDR with high realism, the attack can also modify the generated texture, compared to low realism.

# H  Computational Resources

The experiments were conducted on an internal cluster equipped with RTX 3090s and RTX 2080 TIs. In total, we have logged almost 5000 GPU hours for the experiments and testing. Most of the compute was spent on exploration and the diffusion experiments, with over 2000 hours being spent on the latter alone.

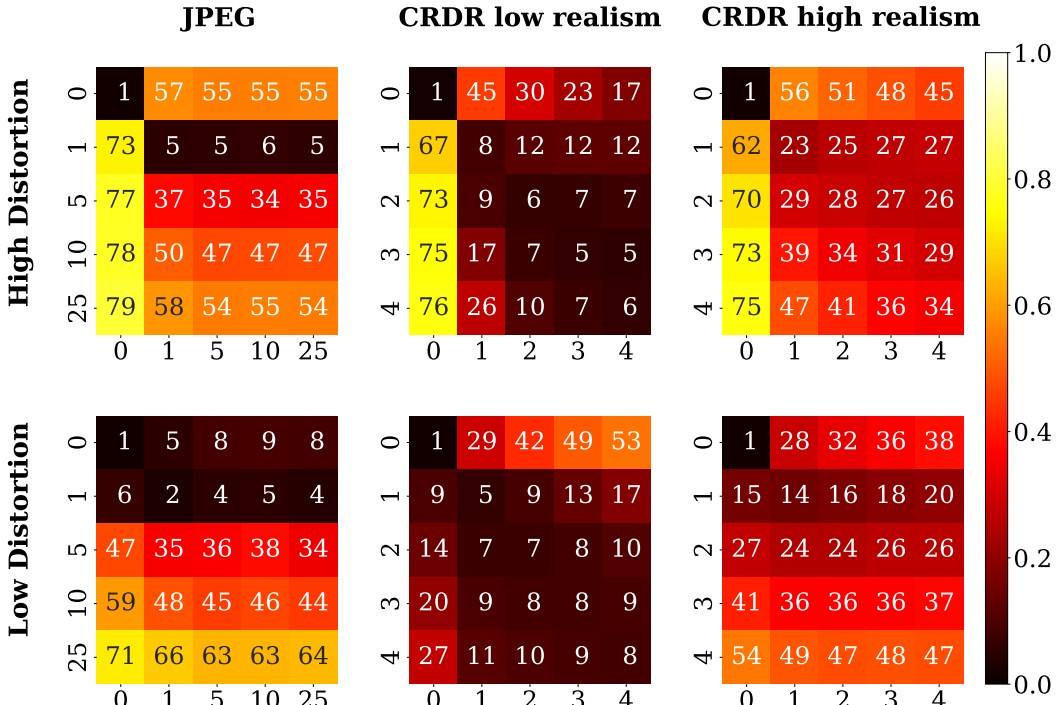

Figure 7: Robust accuracy (%) under iterative defenses and PGD attacks with $\epsilon$ 8/255. Rows indicate the number of attack iterations; columns indicate the number of defense iterations. Darker colors represent lower robust accuracy. Increasing the number of JPEG defense iterations leads primarily to gradient masking. In contrast, CRDR shows modest gains from iterative defense under weaker attacks.

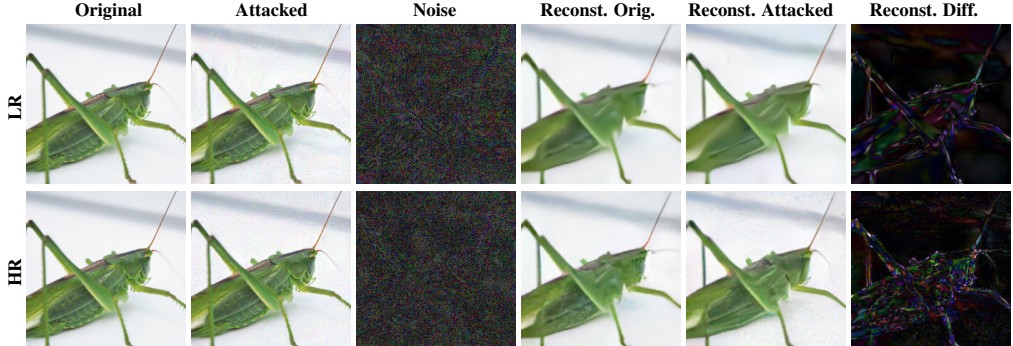

Figure 8: Comparison of original and attacked images, their differences, and reconstructions. CRDR with low realism and high realism, original, attacked, adversarial noise, reconstructed, reconstructed attacked, and the difference between the two reconstructed. For better visualisation, the magnitude of the adversarial noise and reconstructed difference is multiplied by 10 and 3, respectively. We used our default PGD attack with $\epsilon = 8/255$ and $n = 10$ iterations.

