# OpenReview forum: "Keep It Real: Challenges in Attacking Compression-Based Adversarial Purification"
_NeurIPS.cc/2025/Workshop/Reliable_ML — NeurIPS 2025 - Reliable ML Workshop_

### Official Review · Reviewer_C12o · 2025-09-08
**Simulation study of ``realism'' against adversarial attacks**

**Rating:** 7
**Confidence:** 2

**Review:**

Summary: The authors focus on an approach to maintaining classifier performance against adversarial attacks, noticeably through ``realism''. This realism is achieved through playing around with hyper-parameters in the loss function for compression models. The authors then compare various different defenses against different kinds of attacks; the results support their claim of the benefits of realism.

Strengths: The simulation results are quite thorough, making a decent empirical case for the claim that the authors are making. Writing is relatively clear.

Weakness: There's little to no intuition as for why this realism works for these kinds of defenses. The authors try and justify it through minimizing data points that are too ``out-of-distribution'', but this is somewhat lacking. As a reader, its hard for me to tell whether this is a systematic pattern, or just something that happened in this specific example.

Suggestions: The tables presented are quite unclear to me; they probably need to be labeled with the units (i.e. percentage I assume). It would also probably be helpful to include standard errors for readers to gauge the spread.

---

### Official Review · Reviewer_MTfo · 2025-09-15
**Empirical results on compression-based defenses against adversarial perturbations**

**Rating:** 5
**Confidence:** 3

**Review:**

# Summary

The paper investigates the compression-based defense against adversarial perturbations in image classification. Such defense will compress and then reconstruct the input image before giving it to the classifier, aiming to remove the adversarial noise while preserving the semantic contents during compression. The paper argues that realism is the key to robustness by evaluating several different attacks against compression-based defenses. The experiment shows that compression-based defenses with high realism outperform the others, even under adaptive attacks.

# Strengths

The writing is clear and the visualization helps understanding the arguments.

The paper isolates realism as the critical factor for robustness, a contribution not systematically studied before. The experiments include extensive attacks and compression-based defenses. Specifically, the authors include a “Hyperprior Noise” variant to rule out the robustness resulting from gradient masking, instead of realism.

# Weaknesses
As the authors have mentioned, previous works have argued that human-aligned compression contributes to robustness.
And the paper is only verifying that realism contributes to robustness, while one would expect the author could argue some intrinsic obstacle for the attacks against compression-based defenses (for example the authors explain that high-realism defenses will project adversarial inputs back onto the manifold of natural images but they do not present quantitative evidences).

Also, the results focus on image classification, without exploring the other domains. For example, how do compression schemes for audio/video/text with (properly defined) high realism help with robustness against adversarial attacks?

# Suggestions

In line 195, the gradient of $L(f(x), y)$ w.r.t. x should not be denoted by $\nabla_x f$, which is the gradient of $f$.

It will be good to define robust accuracy, as it is the important metric in the tables.